# Modeling the Mechanobiology of Cancer Cell Migration Using 3D Biomimetic Hydrogels

**DOI:** 10.3390/gels7010017

**Published:** 2021-02-12

**Authors:** Xabier Morales, Iván Cortés-Domínguez, Carlos Ortiz-de-Solorzano

**Affiliations:** IDISNA, Ciberonc and Solid Tumors and Biomarkers Program, Center for Applied Medical Research, University of Navarra, 31008 Pamplona, Spain; xmorales@unav.es (X.M.); icortesd@unav.es (I.C.-D.)

**Keywords:** hydrogel, collagen, Matrigel, extracellular matrix, mechanobiology, amoeboid-mesenchymal transition, cancer, cell migration, microfluidic devices, bioprinting

## Abstract

Understanding how cancer cells migrate, and how this migration is affected by the mechanical and chemical composition of the extracellular matrix (ECM) is critical to investigate and possibly interfere with the metastatic process, which is responsible for most cancer-related deaths. In this article we review the state of the art about the use of hydrogel-based three-dimensional (3D) scaffolds as artificial platforms to model the mechanobiology of cancer cell migration. We start by briefly reviewing the concept and composition of the extracellular matrix (ECM) and the materials commonly used to recreate the cancerous ECM. Then we summarize the most relevant knowledge about the mechanobiology of cancer cell migration that has been obtained using 3D hydrogel scaffolds, and relate those discoveries to what has been observed in the clinical management of solid tumors. Finally, we review some recent methodological developments, specifically the use of novel bioprinting techniques and microfluidics to create realistic hydrogel-based models of the cancer ECM, and some of their applications in the context of the study of cancer cell migration.

## 1. Introduction

Cell migration is crucial for several physiological processes as diverse as tissue morphogenesis, immune cell trafficking, wound repair, and metastasis, one of the hallmarks of cancer malignancy [1,2]. Cell assays based on two-dimensional (2D) cellular models, such as wound healing or scratch-based assays, are still widely used for migration research. Therefore, most basic concepts about cell migration have been described from the study of cell motility on top of 2D substrates made of one or several extracellular matrix (ECM) components [3,4]. In particular, the effect of relevant environmental factors, such as the ECM composition, the diffusion of chemical factors, or the topology and mechanical properties of the substrate in how cells migrate has been mostly analyzed in 2D [5] even if 2D systems cannot faithfully recapitulate the molecular and biomechanical complexity of 3D in vivo environments. Indeed, there are specific characteristics of 3D environments that 2D models are not able to replicate, such as the cell’s spatial confinement, or cell–cell and cell–matrix interactions that affect proliferation, differentiation or the response to migration stimuli [6,7]. These limitations of 2D cellular models have fostered the development of hydrogel-based 3D cellular models that more faithfully replicate the native environment of migrating cells.

In this review, we start by reminding the reader of the concept and principal elements of the extracellular matrix of tissues. Then we review the state of the art on the materials commonly used to fabricate hydrogels that mimic the composition and architecture of normal and cancer tissues, and in that context, review the latest research on the mechanobiology of 3D cancer cell migration. Finally, we present the latest engineering developments in bioprinting and the use of microfluidic devices to create realistic 3D environments to efficiently study cancer cell migration.

## 2. The Extracellular Matrix (ECM)

The extracellular matrix is a highly organized protein structure that provides biochemical homeostasis and structural support to cells, tissues, and organs. The ECM is made of a complex network of cell-secreted macromolecules, including fibrous proteins, glycoproteins, and proteoglycans (PGs) [8,9] (Figure 1). The relative abundance and spatial organization of these ECM constituents confer each tissue type with unique physical and biochemical properties [5,10]. These properties, e.g., the rigidity of the matrix or its porosity, affect cell behavior and actively contribute to homeostasis and tissue disease [11].

Collagen is the major structural constituent of the ECM. Collagen fibers self-organize into 3D networks whose density and tensile strength play a key role in cell migration and adhesion, affecting the maintenance of normal tissue physiology or the onset of pathological tissue behavior [10,12]. For instance, in the context of cancer research, the aberrant expression, deposition, alignment or cross-linking of various collagen subtypes have been associated with the occurrence of mesenchymal–epithelial transition (MET), tumor dissemination, or drug resistance [13].

Non-collagenous glycoproteins, such as fibronectin, laminin, and elastin, are adhesive proteins extensively expressed in the ECM. These proteins, interspersed between the collagen fibers, increase the structural integrity of the mesh and participate in cell–matrix interactions and cell signaling, through binding with integrins and other cell surface receptors, such as syndecans or a bulky glycocalyx [14,15].

PGs and glycosaminoglycans (GAGs) form another relevant family of the ECM network. PGs form a gel-like, amorphous material that fills the interstitial spaces of the ECM, embedding other scaffolding proteins. Both PGs and GAGs constitute also a reservoir of bioactive molecules and growth factors, e.g., TGF-β or EGF, that regulate fundamental cell processes such as cell proliferation, migration, and differentiation [16].

Bearing in mind the complexity of the ECM composition, it is not surprising that alterations in one or a few constituents of the ECM may elicit remarkable changes in its biochemical and physical properties, leading to dysregulated cell behavior and disease. Accordingly, in vitro studies of disease must, to the largest possible extent, be based in models that replicate the biomechanical properties and the composition of the target ECM, in order to be of biological significance.

## 3. Three-Dimensional (3D) ECM-Mimicking Scaffolds: Materials

Two-dimensional (2D) cellular models are being replaced by 3D hydrogel-based cell culture models [17,18]. Hydrogels are highly hydrated 3D polymeric scaffolds made of one or several physically or chemically cross-linked ECM-derived proteins: collagen derivatives, glycoproteins, polysaccharides such as hyaluronic acid (HA), or reconstituted cell-derived matrices (CDM) such as Matrigel^©^ (Corning) [19,20]. The biological origin of these materials endows hydrogels with inherent cytocompatibility, adhesion, and excellent remodeling properties, making them ideal to study the mechanisms of tumor initiation, invasion, metastasis, or drug resistance [7,17,21,22].

Collagen is the main structural component of the ECM. Among over 30 subtypes, type I collagen is the most frequent one found in connective tissues [23]. Due to their structural properties, biocompatibility, permeability and degradability, collagen-based hydrogels are ideal to replicate tissue development and regeneration, as well as tumor biology [24,25]. For instance, type I collagen hydrogels have been extensively used as supportive scaffolds for growing spheroids, organoids, and cancer stem cells to study the tumor niche or the efficacy of anticancer drugs [26,27].

The polymerization properties of collagen are highly dependent on its concentration, temperature, pH, ionic forces, and level of cross-linking [28]. Similarly, the source of the protein, e.g., rat-tail tendon, porcine skin, or bovine skin, has an impact on the gelation kinetics, as well as on the mechanical properties and structure of the fibers [29]. Therefore, by properly tuning the polymerization parameters, changes can be effected in the topology and mechanical properties of the scaffolds, that affect the behavior of cells embedded in them [30]. For instance, by adjusting the ionic strength with NaCl, balancing the pH near the isoelectric point, or increasing the gelation temperature, collagen hydrogels can be fabricated with tunable optical and mechanical properties [31]. Furthermore, collagen fiber packing increases the stability and rigidity of the scaffold, which in turn affects the migration mode and speed of the cells [32]. Likewise, high collagen concentration renders matrices with small pores, reduced elasticity and compressive behavior, which in turn lowers the speed and may force a phenotypic switch of migrating cells [33].

Collagen hydrogels exhibit plastic compression behavior and viscoelastic properties, with a non-linear strain stiffening effect [28]. Mechanically, the balance between the hydrogel’s elastic and compression moduli determines its stiffness, which in turn affects cellular fates. Indeed, stromal stiffening has been associated with malignant tumor progression, drug resistance, and increased tumor-induced angiogenesis [34]. Similarly, highly cross-linked collagen has been associated with poor cancer prognosis, and the aberrant expression of the lysyl oxidase family (LOX) induces collagen condensation and has been related to integrin-mediated cell proliferation and invasiveness in various types of carcinoma [35]. However, excessive collagen cross-linking negatively regulates the invasion machinery as hydrogels become less susceptible to cell degradation and remodeling [36].

Dynamic remodeling of the stromal collagen network is essential for the development of normal tissues and is also one of the hallmarks of tumor progression [37]. Remodeling involves deposition, compaction, and aberrant synthesis of cell-secreted fibers in the tumor surrounding ECM, as well as the upregulation of matrix degradation [38]. Most tissue repair and regeneration strategies require high rates of collagen degradation and/or remodeling [39]. To this end, collagen hydrogels contain inherent proteolysis-sensitive sites available for the activity of cell-secreted collagenases and enzymes, such as matrix metalloproteinases (MMPs) [40]. Furthermore, collagen fibers exhibit interspersed integrin-binding arginylglycylaspartic acid (RGD) domains that promote cell-matrix attachment. Through these binding sites, cells can exert tensile forces that align and pack the collagen fibers. It is noteworthy that collagen anisotropy has been widely described surrounding tumor spheroids and tumor cells in vitro and has been associated with malignancy [41]. In summary, due to their biological origin, high plasticity, outstanding biocompatibility and biomechanical properties, collagen hydrogels are ideal to study cancer cell migration [42,43].

Gelatin hydrogels are a subtype of collagen scaffolds resulting from partial physical, chemical, or enzymatic hydrolysis of collagen, which can be isolated from bovine or porcine skin, bones, ligaments, and tendons, or from fish scales [44]. Due to its diverse sources and extraction processes, gelatin hydrogels exhibit variable gelation and mechanical properties. Similarly, temperature, ionic strength, pH, or concentration also influence gelatin behavior. For instance, hydrogels can form spontaneously upon cooling the gelatin solution, in a sol-gel transition, or through physical and chemical modifications [45]. The denaturation of collagen confers gelatin with similar chemical and mechanical properties to those of the related collagen scaffolds. Indeed, gelatin contains intrinsic RGD motifs for cell adhesion and MMP-cleavable sequences that support cell functions. Furthermore, gelatin hydrogels display high retaining affinity for soluble growth factors [46]. Altogether, gelatin hydrogels have received significant attention from the biomedical field because of their low cost, low antigenicity, and outstanding biocompatibility. However, these hydrogels display low degradation rates, high viscosity, poor handling properties, and lack thermal and mechanical stability, which limits their applications [47]. To address this, gelatin has been synergistically conjugated with polysaccharides such as chitosan, alginate, or HA, producing scaffolds with mechanical properties that are appropriate for biomedical applications [48]. Furthermore, chemical modifications of gelatin hold the ability to tune the stretching and mechanical strains of the scaffolds. Among these, low-cytotoxicity photo-cross-linking of methacrylamide residues produces gelatin scaffolds (Gel-MA) with tunable stiffness, that can be used as realistic models of tumor invasiveness, as well as appealing platforms for cancer-targeted drug delivery [49,50].

Hyaluronic acid is a natural, linear, endogenous polysaccharide, widely represented in connective tissues and the tumor stroma. HA plays essential physiological and biological roles in cell migration, proliferation, morphogenesis, and inflammatory diseases [51]. Due to their properties, i.e., cytocompatibility, biodegradability, viscoelasticity, and engrafting ability, HA-based hydrogels have demonstrated great potential for mimicking 3D tissue environments [52]. The design of these hydrogels requires careful choice of several parameters -e.g., HA source, molecular weight, buffer solution, crosslinking source and reticulation degree- [52]. Commonly, HA is obtained from animal tissues e.g., bovine vitreous humor, sharkskin, or human umbilical cord. Due to their native origin, these polymers contain endotoxins and residual proteins that can trigger the immune response thus hindering their biomedical application. Consequently, alternative techniques have been developed in recent years to isolate HA polymers from bacteria or algae with barely detectable levels of immunogenic proteins [53].

HA hydrogels can be classified as “physical” or “chemical” gels depending on the crosslinking reaction. Physical HA hydrogels are crosslinked by heating or cooling the HA solution, while chemical HA hydrogels are crosslinked using chemical or radiation-based treatments [52]. By modulating the cross-linking degree of the lattice, HA-based hydrogels can be finely tuned to produce matrices with highly controlled mechanical properties. For instance, HA scaffolds have been used to generate soft tensile environments that mimic the stiffness of breast, lung or brain tumors [54,55]. Interestingly, increased deposition of HA has been reported in the stroma of several types of solid tumors, including lung or breast tumors. Furthermore, the deregulated fragmentation of HA fibers by tumor cell-mediated reactive oxygen species (ROS) release triggers pro-invasive cues in a positive feedback loop [56,57]. Similarly, aberrant HA expression promotes tumorigenesis and malignant progression in breast cancer, where it is also associated with poor prognosis [58]. In particular, HA polysaccharides promote tumor spreading by offering linking sites to embedded cells through several transmembrane receptors such as CD44 and ICAM-1, overexpressed in many solid tumors. Consequently, HA-based hydrogels constitute excellent platforms for the design of antitumor drug delivery strategies targeting CD44 peptides [59]. Furthermore, the HA network stands out for its viscoelastic and cushioning properties, essential for the proper functioning of bone and articular cartilage. Accordingly, several studies have reported how HA hydrogels induce the osteogenic differentiation of mesenchymal stem cells into chondroblasts and osteoblasts that help repair and regenerate soft and hard tissues [60]. Furthermore, HA lattices display a porous framework that is optimal for cell implantation and ingrowth in cartilage and bone tissue engineering, as well as tumor immunotherapeutic drug delivery [61,62].

Despite this mechanical and biologic versatility, unmodified HA hydrogels cannot faithfully reproduce tissue remodeling events, as these hydrogels are insensitive to cell-secreted proteolytic peptides and block the expression of various MMPs by modulating the MAPK kinases signaling pathway, inhibiting the secretion of pro-inflammatory molecules, such as interleukin-6 [63]. Consequently, HA scaffolds are often conjugated with RGD motifs for integrin binding, and with peptides susceptible to degradation by MMPs in order to promote tumor cell invasive behavior [64]. These modifications however, alter the architecture of the network and introduce a high level of uncertainty and less controllable experimental conditions. Finally, the biological instability of native HA requires the use of chemical reagents for network cross-linking, such as glutaraldehyde, which can elicit potential cytotoxic effects [65].

Other chemically cross-linked polysaccharides such as alginates have also been used in biomedical applications. In particular, alginate scaffolds show a pH-dependent viscosity, which endows these polymers with a controllable structure and chelating behavior that favors drug immobilization for cancer-targeted therapies, as well as for tissue regeneration strategies [66].

In the previous paragraphs we have described 3D hydrogels made of purified matrix proteins. These single-element biomaterial scaffolds are far removed from the complex composition of real tissues. Due to their native origin, cell-derived matrices (CDM) offer a more physiological alternative when studying in vivo complex cell–matrix interactions [67]. Due to its particular biophysical properties, Matrigel is overwhelmingly the most used CDM in cancer studies [27,68]. Matrigel is a commercially available complex derived from the basement membrane (BM) of the Engelbreth–Holm–Swarm (EHS) mouse sarcoma. It is mostly composed of laminin, collagen IV and entactin. Besides these structural components, Matrigel also contains abundant growth factors including EGF, TGF-β and PDGF, and other matrix proteins that are mediators of cell growth, differentiation and self-organization into 3D structures [69]. Noteworthily, it has been reported that this abundance of soluble cues within Matrigel contributes to the differentiation of cancer stem cells in several tumor types [70].

Matrigel polymerizes at physiological temperature, displays linear elastic properties and behaves mostly as a soft material [71]. Matrigel-based hydrogels are highly sensitive to MMPs-mediated remodeling, as required for tumor progression [42,72]. However, its linear elastic behavior results in complex handling and poorly controlled mechanical properties. Furthermore, Matrigel-based studies are not easy to translate to the clinic due to its murine origin and high antigenicity. Finally, Matrigel contains some elements that are not extensively contained in the tumor stroma, and show high variability between batches, complicating the reproducibility of cell culture experiments [70]. To address some of these issues, Matrigel is often cross-linked with collagen or synthetic materials such as polyethylene glycol (PEG) [73]. These mixed Matrigel hydrogels display a dense, poorly organized architecture that yields highly heterogeneous scaffolds that closely resemble the disorganized BM at the leading edge of tumor invasion. Interestingly, in these scaffolds both pore size and lattice stiffness increase with the concentration of Matrigel, facilitating cell tractions and migration through these matrices [74].

Over the last decade, (semi)synthetic-based hydrogels have been developed that are well-suited for in vitro 3D biomimetic cell-culture. Among them, PEG, polylactic acid (PLA) or poly(lactic-co-glycolic acid) (PLGA) have been used to model the tumor ECM [75]. These polymers, unlike native biomaterials, produce versatile scaffolds with highly tunable physicochemical properties, such as hydrogel stiffness and/or density [76]. However, they fail to mimic the complexity of natural tissues, and display minimal or no intrinsic bioactivity. Indeed, these polymers do not exhibit functional ligands, requiring either chemical insertion of MMP-sensitive peptides and integrin-binding domains (RGD motifs) or cross-linking with native proteins [77]. Alternatively, mixing synthetic-based hydrogels with Matrigel, collagen, or collagen-derivatives overcomes these limitations and improves the mechanical properties of the scaffolds [78]. These hybrid hydrogels constitute ideal scaffolds for tissue engineering due to their easy-tunable, low-cost, and high-reproducibility properties. Similarly, synthetic polymers based on silicones such as polydimethylsiloxane (PDMS) are increasingly being used as 3D bio-scaffolds. PDMS scaffolds display outstanding biocompatibility, low antigenicity, and a non-degradable composition that makes them an excellent support for cell cultures. Indeed, its well-defined roughness and porosity favor the cell–cell and cell–matrix interactions that are required for cell homing and engrafting in cell therapies [79]. Furthermore, the combination of PDMS scaffolds with microfluidic techniques enables the miniaturization of highly complex biological processes, e.g., tumor invasion, metastasis, or angiogenic processes, allowing rapid analysis, excellent reproducibility, and low reagent demand, thus minimizing experimental costs [80].

In summary, due to their fine-tunable biophysical properties and biocompatibility, 3D hydrogels are outstanding platforms where to gain insights into cell–matrix interactions and cell behavior within normal tissues and in the tumor-adjacent stroma. To this end, a properly designed scaffold must find an equilibrium between the use of a bare, highly controllable mechanical and morphological environment or else, a biologically relevant but poorly controlled morpho-mechanical environment. Indeed, “simplified” hydrogels made of one ECM-protein, allow optimal control of the experimental, biomechanical conditions, but compromise the relevance of the results, since their simplified composition does not accurately reflect the natural complexity of the ECM. Alternatively, the use of mixed matrices of complex composition, or CDMs, is more appropriate to recapitulate cell–matrix interactions, but allows limited control of the biomechanical properties of the hydrogel, thus complicating the isolation of the effect of a specific ECM element, and reducing the reproducibility and comparability between studies.

## 4. Three-Dimensional (3D) ECM-Mimicking Scaffolds: Sensing, Signaling and Remodeling

The environmental factors that affect how cells self-organize, proliferate and migrate within their 3D extracellular environment can be subdivided into mechanical (elasticity and rigidity of the substrate), geometrical (topology, anisotropy and confinement), chemical (adhesive ligands, bioactive cues, and substrate degradability), and structural (composition and cross-linking of ECM constituents) (Figure 2) [81]. The way cells probe and adjust to these biophysical cues largely determines the behavior of the cells and tissues [82].

Cells probe the spatial organization of tissues via their membrane receptors, e.g., integrins. Through this process, called mechanosensing, cells become “aware” of the chemical and physical properties of the surrounding ECM, and “feel” tensile forces acting on the substrate (Figure 3) [83]. Growing tumors dynamically modify the composition, density, and architecture of the surrounding ECM. The tumor cells sense these changes, and adapt their biomechanical machinery to the new ECM properties [81,84]. In particular, ECM sensing drives the directed migration of tumor cells away from the primary niche and towards healthy tissues, a phenomenon known as contact guidance. During this process, tumor cells modulate the actin cytoskeleton dynamics and establish specific adhesive structures in response to mechanical and chemotactic cues (Figure 3) [85,86]. Indeed, the actin cytoskeleton plays a critical role in ECM mechanosensing, as cells extend actin-based protrusions to probe and attach to ECM ligands-lamellipodia, pseudopods or filopodia, or to degrade the surrounding matrix, invadopodia. These protrusions generate tensile and compressing forces aimed at adapting the cell’s morphology to the substrate stiffness and to navigate through the ECM [87]. The transduction of these actin-dependent tensile forces to the ECM occurs via integrin-dependent adhesion sites, termed focal adhesions (FAs), which are specialized anchoring sites located on the plasma membrane, providing a nexus between the actomyosin contractile machinery and the ECM [88,89]. These multiprotein clutches are composed of a core of integrins that actively participate in the recognition and anchoring to specific ECM ligands. Thereby, FAs sense local changes in the ECM that shift integrins to a high-affinity adhesive state in a stiffness-dependent manner (Figure 3) [90].

In rigid three-dimensional substrates, the cytoskeleton of migrating cells becomes polarized, in a process that involves actin stress fiber formation, expression of integrins, and strengthening of FAs via the activation of the Rho/ Rho-associated protein kinase (ROCK) pathway [89,91]. The high substrate stiffness upregulates myosin-II expression and actomyosin fiber contractility, leading to elongated pro-migratory cell shapes. In soft substrates, however, a non-polarized cytoskeleton and short-lived actin-rich cell surface protrusions, typical of low migrating cells, are commonly observed (Figure 3) [92,93,94]. Consistent with these observations, tumor malignancy is commonly associated with stromal rigidity and densification, and it has been described that Rho/ROCK signaling enhances tumor spread by promoting the alignment of stress fibers and actomyosin-dependent cell contractility through Arp2/3 and MLC proteins (Figure 3) [95]. Indeed, both high-stiffness and high-density substrates trigger specialized types of directed cell migration named durotaxis and haptotaxis. These particular forms of guided locomotion require efficient cell spread towards increasing gradients of extracellular stiffness and density, respectively [96,97]. This has been widely described during neoplastic progression both in the tumor vasculature and in the surrounding stroma [98]. Under these biomechanical conditions, tumor cells exhibit increased integrin clustering, upregulated FA dynamics, enhanced cytoskeletal tensile forces, altered growth and invasive behavior. Several groups have described how these events are tightly controlled by phosphorylation of the focal adhesion kinase (FAK)/Src signaling pathway (Figure 3) [99,100]. Indeed, FAK expression is negatively regulated in soft matrices regardless of their molecular composition, indicating that mechanical inhibition of FAK leads to weak FAK–integrin–matrix interactions. Conversely, both aberrant FAK phosphorylation and β1 integrin expression have been specifically associated with stromal rigidity and poor prognosis in various types of solid tumors [101,102,103]. Interesting in this context, FAK signaling determines a host of cellular responses, including cell proliferation, differentiation and migration in a stiffness-dependent manner through FAs assembly (Figure 3) [104]. A range of molecular factors have been associated with FAs turnover and maturation. However, ECM stiffness and actomyosin-dependent contractility also play a pivotal role in FAs assembly, via coupling of the scaffold proteins vinculin and talin to integrins [105]. Accordingly, in soft matrices, weak RhoA-driven cytoskeletal tensile forces fail to stabilize cell adhesions, rendering non-effective cell spread. This negatively regulates the recruitment of integrins and induces integrin endocytosis and subsequent lysosomal degradation (Figure 3) [102,106]. Therefore, ECM stiffness and densification modulate the size and number of FAs of cells immersed in 3D matrices. It is noteworthy that hydrogels with a high Young’s modulus (G’) display large numbers of FA-preforming vinculin and paxillin clusters [107,108]. This vinculin overexpression is closely associated with both increased recruitment of β1 and β3 integrins subtypes and strong tensile forces indicative of super-mature FAs. Recent works have described that these contractile and tensile forces are higher in Matrigel–collagen composite hydrogels than in pure collagen scaffolds, due to greater lattice rigidity and density of ligands [42]. Similarly, high level of collagen cross-linking by LOX overexpression promotes integrin recruitment, FAs assembly, and strong tensile forces that trigger the oncogenic transformation and tumor dissemination [109]. Conversely, soft substrates may affect vinculin recruitment, leading to FAs disassembly concomitant with disrupted cytoskeletal-mediated stress mechanotransduction and impaired tensile forces [107].

Consistent with these studies, some recent works have described that the spatiotemporal maturation of FAs does not depend exclusively on molecular processes, but is also largely controlled by topographic cues, as cells can distinguish and adapt the specific rates of FAs growth to different micro and nanoscale roughness patterns of the substrate (e.g., microgrooves, ridges, or pillars) [110]. For instance, substrates enriched in ridge and groove patterns cause actin cytoskeleton rearrangement and integrin crowding along these structures, enhancing FAs maturation and cell guidance via the Rho/ROCK pathway [111]. The polarization of these FAs induces tumor cell dissemination through collagen fibers and causes local alignment and densification of the ECM in the direction of the underlying forces [112,113]. Indeed, tumor cells can remodel rigid substrates using RhoA-integrin-mediated cytoskeletal tensile forces, to form highly aligned and packed fibrotic areas perpendicular to the boundary of a tumor, facilitating cell invasion. These anisotropic matrices enhance persistent cell migration of various types of tumor cells along its densely packed fibers, facilitating cancer progression [113,114]. It is noteworthy that local ECM anisotropy and fiber packing are more prevalent in hydrogels of mixed composition than in collagen-only scaffolds, where it is associated with higher migration integrin-dependent rates [42].

Matrix stiffness also modulates the expression of mechanosensitive genes through FAK signaling [115]. For instance, the clumping of actin stress fibers elicits nuclear translocation of the YAP/TAZ cytoplasmic complex. Once in the nucleus, these proteins act as transcriptional co-activators of genes involved in the regulation of cytoskeleton and FA constituents (Figure 3) [116]. Accordingly, in what seems to be a clear feedforward positive loop, in various types of solid tumor, YAP causes matrix stiffness and fiber densification by increasing stromal deposition of collagen [117]. During this process, it has been described how tumor cells orchestrate the recruitment of cancer-associated fibroblasts (CAFs) in a paracrine manner, by releasing pro-fibrotic factors such as TGF-β, PDGF, or EGF [118]. CAFs are the major depositors of ECM components in the tumor environment. They actively participate in the stromal, LOX-dependent fibrotic process called desmoplasia. Consistently, an altered balance in the cellular stromal deposition of collagen, laminin or HA is commonly associated with tumor progression and poor prognosis in most cancer types [119,120].

The tumor environment undergoes dramatic changes during the tumorigenesis process leading to a perturbed, poorly organized architecture, consequence of altered levels of collagen deposition, cross-linking, and/or fiber alignment. For instance, excessive CAFs collagen deposition in the tumor-adjacent stroma is associated with a dramatic reduction in the matrix pore size that causes high cell confinement, and hinders cell motility, forcing moving cells to degrade the surrounding matrix via proteolytic enzymes [33]. Similarly, invasive phenotypes have been described in LOX-dependent cross-linked collagen hydrogels. These highly cross-linked matrices promote tumor spread via slow β1 integrin-FAK/Src-mediated locomotion in combination with high matrix degradation rates [121]. In contrast, tumor cells immersed in heterogeneous, larger pore-sized gels made of a mixture of collagen and Matrigel exhibit high migration rates, as cell easily navigate through the underlying collagen mesh [42]. Furthermore, ECM porosity determines the permeability of the matrix, as well as the establishment of biochemical gradients that allow cell guidance. The small pore sizes observed in the tumor stroma limits the diffusion of soluble growth factors and oxygen through the ECM. These hypoxic events induce cell metabolic reprogramming and cause an acidic environment that is associated with increased metastasis and high remodeling rates of the tumor-adjacent stroma [122,123].

In response to limiting soluble factors and high cell confinement, tumor cells increase proteolytic-based matrix remodeling, leading to the breakdown of the BM that facilitates angiogenesis and metastasis [124]. This cell degradation machinery includes a wide range of soluble proteases, such as the family of MMPs, A disintegrin and metalloproteinases (ADAMs), or cathepsins, which are released into the pericellular space in association with surface receptors, such as integrins [125,126]. Indeed, recent studies have described that MMP-based proteolytic activity in collagen-based hydrogels regulates stiffness-dependent invasion in various cancer cell types, in association with the β1 and β3 integrins [127,128]. Similarly, increased collagen cross-linking in LOX high-expression tumors causes the upregulation of these proteases, facilitating tumor progression by enlarging matrix pore diameters through MMP-dependent ECM cleavage [129,130]. On the other hand, the increased presence of micro and nano-patterns in the tumor stroma favors the recruitment and stabilization of invadopodia in the cell membrane via integrin-dependent signaling. These actin-rich membrane protrusions drive local degradation of ECM through the secretion of vesicles filled with proteolytic material, thus increasing the invasive potential of tumor cells [36]. Local degradation of the ECM often results in the formation of microtracks within the collagen matrix. These tube-like structures allow collective directed cell migration along paths of least resistance, contributing to increased metastasis and tumor dissemination toward blood vessels [131]. Intravital microscopy has revealed how a “leader cell” creates small proteolytic microchannels, being followed by other accompanying cells, in a cell-cell contact-dependent manner. This collective cell migration results in an aggressive and coordinated mode of invasion, as it minimizes the energy requirements for tumor cells, and confers decreased sensitivity to chemotherapeutic agents [132]. Furthermore, proteolytic cleavage of ECM fibers is essential for developing guidance cues gradients. In particular, the MMP-dependent activity releases growth factors, such as TGF-β or VEGF, immobilized within the tumor stroma that remain inactive while still anchored to PGs or CDM-type matrices. Consequently, the release of these bioactive cues creates chemotactic gradients that drive tumor growth, angiogenesis, and metastasis processes [133].

In summary, a delicate balance between ECM stiffness and lattice confinement, i.e., porosity, density, or cross-linking, exists that finely orchestrates a host of cellular responses to tailor mechanobiology machinery to the matrix properties. Consequently, the dysregulation of one of these variables fatally alters cell behavior, triggering cellular malignancy and enhancing the ECM remodeling strategies and tumor dissemination.

## 5. ECM Mechanobiological Stimuli Govern Cell Migration Plasticity within 3D Environments

The highly disorganized tumor ECM presents moving cells with a variety of physical elements that can either facilitate or oppose their motility. Consequently, tumor cells exhibit extensive mechanobiological plasticity to adapt their migratory machinery to the physical properties of the ECM, generating well-differentiated locomotion strategies.

Relevant examples of 2D migration have been described in mammalian tissues, including the movement of epithelial cells during wound closure or the patrolling of immune cells in the inflammation response [134,135]. Furthermore, the basic concepts on cell migration have been traditionally defined from the analysis of isolated cells during their locomotion on top of 2D planar substrates [3]. Under these conditions, cells move predominantly using lamellipodia-based locomotion. This is characterized by a multi-step cycle of membrane extension, i.e., fan-shaped protrusion, adhesion, stabilization at the leading edge, and generation of traction forces through actomyosin cables, followed by translocation of the cell body [136]. During these events, the small GTPase family, i.e., Rac1, Cdc42, and RhoA kinases, is spatially and temporally activated at the leading edge of the cell, coordinating actin-based membrane protrusion, cell adhesion, and polarization [137]. Even though this mechanobiological procedure is highly conserved in 3D, the dimensionality and intricate composition of 3D tissues forces moving cells to adopt specific strategies to navigate within 3D matrices [81]. Specifically, cells might either locally degrade the pericellular ECM or dynamically adapt their intracellular tension to crawl through ECM pores. These are the two principal modes of 3D migration, termed mesenchymal and amoeboid, respectively (Figure 4) [138]. The interplay between RhoA and Rac1 signaling pathways potentially controls the mechanosensitivity of the matrix and tilts 3D migration towards one or the other phenotype [139].

Three–dimensional (3D) mesenchymal tumor cell migration is characterized by elongated spindle-like morphology. Indeed, mesenchymal migrating cells stretch and compress their body to accommodate to the ECM topology. The complex architecture and porosity of 3D environments hamper the apical-to-basal polarity described on 2D substrates, favoring instead a front–rear axis polarity [140]. Furthermore, the formation of fan-shaped protrusions, i.e., lamellipodia, can be physically restricted in highly confined matrices by preventing lateral expansion of the cell membrane [141]. Instead, in mesenchymal 3D migration, actin nucleation drives one or multiple cylindrical membrane protrusions, e.g., pseudopodia or filopodia, that dynamically engage with the ECM (Figure 4) [142]. These processes rely heavily on Rac1 and CdC42 signaling at the leading edge to sustain membrane protrusions and guided cell migration, where RhoA activity is negligible. Instead, FA anchoring and RhoA-directed actomyosin contractility happens at the rear of the cell. Notably, besides promoting nucleation and polarization of actin fibers, Cdc42 is also a critical regulator of cell orientation and persistence through centrosome repositioning of the nucleus [143]. Accordingly, these actin-rich protrusions, might be more relevant for ECM exploring and pathfinding than for ECM degradation. Sustaining this hypothesis, recent studies have shown that tumor cells with disrupted filopodia migrate faster and with higher persistence than tumor cells with intact protrusions, pointing at a mechanosensing role of these structures [144]. Conversely, in amoeboid migration, high RhoA signaling drives increased levels of actomyosin contractility. The effect of these myosin-derived forces on the cell rear sustains high intracellular pressure and membrane tension, leading to the expansion of a single rounded-shaped protrusion or multiple short-lived spherical membrane-blebs at the leading edge of the cell in the direction of cell movement (Figure 4). These structures allow amoeboid-migrating cells to squeeze through pre-existing ECM pores [145]. Amoeboid-based migration commonly exhibits a diffuse pattern of integrins and FAK clusters at the cell surface that yields weak cell-ECM adhesion. Moreover, this migration mode does not rely on matrix degradation, as the cells deform to follow the paths of least resistance, through the ECM pores [42,146]. Mesenchymal migration instead, requires strict cell adhesion for the generation of effective forces, as well as a tunable control over the ECM degradation, especially in tumor cells.

As mentioned in previous paragraphs, the efficient locomotion through 3D environments requires a delicate balance between cell–matrix adhesion and tensile forces that is strictly controlled by the spatiotemporal crosstalk of small GTPases [147]. Indeed, mesenchymal tumor cell migration in complex Matrigel-collagen or in high density single element hydrogels is affected by the biphasic relationship existing between ligand density and cell attachments, which causes the magnitude of traction forces exerted on the substrate not to linearly translate into faster motility. Similarly, soft substrates fail to generate effective tractions forces, thus slowing down the speed of the cell. Therefore, optimal migration rates are observed at intermediate levels of matrix stiffness and cell attachment [42,148]. Likewise, mesenchymal tumor cell speed and persistence are largely controlled by the confinement range of the ECM. For instance, small pore size collagen hydrogels limit cell spread, the nucleus being a limiting factor [149,150]. Hence, mesenchymal cells slow down and can become trapped, forcing pericellular remodeling via MMP activity, generating tube-like paths that allow efficient locomotion in an individual or collective manner (Figure 4) [151]. Conversely, blocking MMPs abrogates mesenchymal cell spread and tensile forces in rigid gels and, ultimately, tumor cell migration and the formation of the metastatic niche [74,127].

Bearing this cell–matrix interdependence in mind, we can conclude that mesenchymal-like migration relies on the creation of new tracks, whereas amoeboid migration relies on path-finding, with minimal interaction with the ECM [152]. Accordingly, due to the slow turnover and biphasic behavior of FAs, as well as the energy requirements derived from the matrix proteolysis, mesenchymal migration is relatively slow (0.1 to 1 µm/min) compared to amoeboid migration (~10 µm/min) [153]. Interestingly, many cancer cells can switch spontaneously from mesenchymal to amoeboid-like migration in low adhesion and high-confinement ECMs, in a process named mesenchymal-amoeboid transition (MAT) (Figure 4). Under these conditions, RhoA-dependent signaling drastically reduces Rac1 activity, suppressing the formation of pseudopods, and triggering the shift to an amoeboid-like phenotype [154]. Similarly, pharmacological inhibition of integrins and MMPs in mesenchymal tumor cells promotes MAT in hydrogels with pores smaller than the cell size, regardless of their composition [74,155,156]. This amoeboid-mesenchymal plasticity exhibited by tumor cells allows more efficient invasion compared to non-adaptive cells, especially in high-confined-hypoxic environments [154].

In this context, the increased ECM remodeling rates reported in mesenchymal-based locomotion locally modulates the elastic behavior of the substrate [157]. Therefore, cells moving within rigid, linear elastic 3D environments, form blunt cylindrical protrusions, termed lobopodia (Figure 4) [136]. This specialized mode of locomotion has been reported exclusively in fibroblasts and tumor cells across CDMs, dermis explants, or highly covalently cross-linked collagen matrices. In fact, the chemically induced loss of cross-linking in CDM-based hydrogels confers non-linear elastic properties to these matrices, triggering the phenotypic transition towards mesenchymal-like modes. Similarly, tumor cells moving in soft, non-linear elastic 3D collagen hydrogels can shift between lobopodial to mesenchymal-based migration [74,158]. Interestingly, lobopodial cells appear to adopt a mixture of both mesenchymal and amoeboid features. Mesenchymal cells can detect the inherent elastic behavior of materials and respond to it by increasing RhoA activity, in order to elicit this phenotypic switch. Namely, during the onset of lobopodia, RhoA-dependent myosin II contractility pushes the nucleus forward like a piston, increasing intracellular pressure [159]. This suggests that the nucleus physically divides the cell into two distinct compartments, the leading edge and the trailing end, maintaining an asymmetric intracellular pressure in between them (Figure 4). As described for amoeboid-like migration, this compartmentalized hydrostatic pressure drives cell motility through small blunt-ended blebs [160]. Furthermore, during lobopodial migration, no polarization of the Rac1, Cdc42, and PI3 kinases is observed as occurs in mesenchymal migration. Despite this, lobopodial cells exhibit robust cell adhesion and cell body polarization unique to mesenchymal locomotion (Figure 4) [161]. In fact, genetic ablation of Rac1 and Cdc42 alters the migration rate of lobopodial tumor cells, indicating that these kinases still influence lobopodia-based migration [139]. Intriguingly, either the knockdown of RhoA or the inhibition of myosin II-dependent contractility triggers a lobopodial-mesenchymal transition without affecting cell speed and persistence [162]. Thus, the choice between these migration phenotypes depends exclusively on the elastic behavior of the ECM and the internal balance of the RhoA-myosin II axis.

In summary, cell locomotion through 3D environments is determined by the mechanical complexity of the tissue, i.e., porosity, lattice compression, or spatial confinement. To overcome these physical barriers, moving cells deploy different strategies displaying different cell–matrix adhesion levels, contractility, type of protrusion, or proteolytic activity, thus giving rise to the mesenchymal, amoeboid, or lobopodial migration modes. The switch between these phenotypes is finely orchestrated by the spatiotemporal crosstalk of the Rho GTPases. Consequently, ECM properties and the Rho pathway govern cell plasticity within 3D environments, favoring efficient navigation by adapting the cell’s biomechanical machinery to each tissue.

## 6. Fabrication of 3D Scaffolds: Microfluidics and Bioprinting for 3D Cancer Migration Assays

As described in previous sections, 3D hydrogel scaffolds are powerful tools to mimic the ECM of tissues, and allow the study of the interactions between cells and the ECM. A careful design and fabrication of the hydrogels is crucial to properly recapitulate specific mechanobiological responses, and in particular, those involved in the migration of normal and cancer cells [163,164]. To this end, the material must be structurally stable during processing and after gelation, adopting the desired 3D geometry [165]. Moreover, hydrogels must display the appropriate mechanical and biochemical properties, i.e., elasticity, porosity, permeability, stiffness, cross-linking, and biodegradability, to render a suitable environment for cell attachment, growth, and differentiation [166,167]. For instance, it has been described that cell migration is impaired in 3D scaffolds, when pores are under a critical value (3 microns) and becomes optimal for pore sizes between 3 and 12 microns [168]. The selection of the gel type and scaffolding technique must allow the creation of navigable hydrogels. However, excessive porosity can compromise the structural viability of the scaffold, and alter the desired geometry or the expected mechanical properties [169,170]. In summary, several properties affect the viability and usability of a scaffold, being key parameters the porosity, mechanical properties, biocompatibility, 3D geometry stability and scaffold resolution [171]. Hence, the selection of a fabrication method depends not only on the capacity to process the material and control the 3D geometries of the fabricated scaffold. It also depends on the scaffold-specific requirements and the materials used [172].

Several scaffold fabrication methods have been reported in the literature. Classical methods such as solvent casting particulate leaching (SCPL) [173,174] or thermally induced phase separation [175,176] are based on the removal of uniformly distributed particles, e.g., salt or other solutes, mixed with the gel, by applying high temperature or particle solvents. Other methods, such as electrospinning, create nanofibrous scaffolds using a polymer that is pumped through a metallic needle or nozzle towards a metallic collector. To this end, a high voltage is applied between needle and collector. When the electrostatic charge in the polymer is larger than the surface tension, a thin filament is ejected towards the collector [177,178]. These classic technologies provide limited control on the scaffold geometry and mechanical properties, due to the difficulty involved in controlling the deposition of the material, particles or solvents [179]. Alternative methods based on rapid prototyping (RP), such as stereolithography [180], fused deposition modeling [181] or selective laser sintering [182] allow more accurate 3D scaffold generation, by relying on a layer by layer fabrication process [183,184]. Table 1 summarizes the advantages, drawbacks, materials and main applications of the principal scaffold fabrication techniques [174,179,185,186,187].

Both classical and RP methods provide limited flexibility when it comes to integrating interstitial flows within the 3D scaffolds, as required to create realistic microenvironments that allow the diffusion of bioactive cues. Microfluidics address this limitation [188], by allowing the combination of hydrogels with micro-structured fluid containing channels. This is done by manufacturing microfluidic devices using biocompatible materials such as PDMS, containing microchannel patterns that can later be filled with the appropriate hydrogels or fluids. For instance, a hydrogel-containing PDMS chamber representing the ECM of a tissue can be “fed” by microchannels that simulate blood or lymphatic vessels. Alternatively, one can directly generate the micropatterns directly from the hydrogel [189].

In this section we focus on two novel fabrication techniques. First, we briefly review bioprinting. This RP technique combines the generation hydrogels of complex, high resolution geometries, while seeding cells on them during the same process. This provides a high degree of flexibility compared to the classical, and other RP methods [190]. Then we focus on microfluidic-based scaffolds. As discussed, these devices allow creating complex multifunctional devices that mimic tissue confinement and recapitulating physiological flows, gradients, permeability and/or crosstalk between different cells or barriers, e.g., the BM or vascular walls [191]. Microfluidic devices can also integrate mechanical or electrical actuators to monitor and trigger bioactive cues [191,192]. Both microfluidics and bioprinting techniques seem especially appropriate to fabricate hydrogel-based scaffolds with controlled properties to mimic the structure and properties of the ECM for migration assays.

### 6.1. Bioprinting

Bioprinting is an emergent RP technique that combines the use of biological materials with classical 3D printable materials to reproduce topologically accurate, functional tissue structures. This technique is based on classic 3D printing technologies, which have proliferated in a variety of areas such as aerospace design, architecture, automobile design, consumer products, electronics, food industry, manufacturing and medical applications [261]. Three-dimensional printing presents several advantages compared to traditional manufacturing techniques. It is a cost-effective method for rapid prototyping, allows the generation of highly complex structures at high resolution, using a high degree of freedom during device conceptualization, with leads to highly customized models [262,263]. Also, printed models are developed under accurate control of the composition and deposition of the material [264].

Three-dimensional printing starts with the design of virtual 3D volumes by computer-aided design (CAD). Three-dimensional printers then physically create these volumes by sequential addition of layers of a printing material, or ink, using deposition techniques such as stereolithography, selective laser sintering (SLS), fused deposition modeling (FDM), powder bed fusion, inkjet printing, extrusion, electrospinning or direct energy deposition (DED) [198]. In this process, the printing material has the same importance as the method itself, since the material must be printable and feature the desired properties [265]. Many metals, polymers, ceramics, composites, or smart materials have been used in 3D printing applications [266,267,268].

Bioprinting inherits the main advantages of 3D printing, for the task of printing cells or biomaterials (bioinks) into biocompatible models with controlled rheological and morphological properties [269]. Different additive methods have been adapted to bioprinting. Inkjet-based bioprinting seeds the bioink as small droplets onto a substrate, using a piezoelectric or thermal actuator. This non-contact technique is very precise but may suffer from material nozzle clogging and thermal and/or mechanical stress on the printed material [270]. Pressure-based methods are broadly used as well, where bioinks are extruded through a nozzle or a needle, in the form of a filament that is layered on the substrate with the desired geometry. This technique allows printing homogeneous bioinks of relative high viscosity, but can compromise cell viability due to the mechanical stress generated during the extrusion process [271]. Laser-based bioprinting follows the stereolitography principle, where a laser source is directed to a photosensitive bioink, curing the desired geometry of the material in a layer-by-layer process. This technique is fast and produces high-resolution printings. However, the use of a laser light source may induce cellular photodamage, and the bioink curing photoinitiators may induce chemical toxicity [272].

As explained, each fabrication technique has specific bioink constraints [273], as they must preserve the mechanical and biochemical properties of the bioink and preserve cell viability while being functional, i.e., printable with the desired 3D structure [274]. Note that not all cell types or cell assays are compatible with all fabrication methods or bioinks. Therefore the most critical aspects of a bioprinting process are the choice the appropriate bioink and setting the optimal printing parameters for the selected technique [275,276,277,278]. Bioink formulations based on cells or cell-biomaterial mixtures have been developed in the last decades, and have been used in a variety of areas such as cancer research [279,280,281], tissue engineering [282,283,284], organ printing [285,286,287] or drug screening [288,289].

Focusing on the fabrication of scaffolds for cell migration assays, bioprinting presents an interesting alternative to other hydrogel fabrication techniques, as it allows creating multifunctional structures through the precise seeding of cells inside of high-resolution bioprinted ECMs. The bioinks used for these assays must have obvious biocompatibility for live cells, sufficient permeability for nutrients and oxygen, defined mechanical properties, e.g., viscoelasticity, stiffness, porosity- while allowing biodegradation and structural stability [290]. Even if they might suffer from some of the already mentioned limitations, i.e., they can be mechanically unstable or must be used combined among others, most of the classical hydrogels reviewed in Section 3 are suitable for bioprinting, as they promote cellular growth, development and proliferation, and display rheological properties that mimic the extracellular microenvironments [80]. Among many others, we describe next a few representative applications of bioprinting in the context of cell migration studies.

Laser-assisted bioprinting procedures (LAB) has been applied by J.M. Bourget et al. to create droplets of cells micropatterned as parallel lines [291]. The droplets, containing human umbilical vein endothelial cells (HUVECs) or HUVECs mixed with bone marrow mesenchymal stem cells (HBMSCs) were microprinted on a substrate layer made of type I rat collagen (Figure 5a). To this end, a pulsed laser excites the upper side of a glass slider that contains a gold absorbing layer. A thin printable solution containing the cells is spread on the bottom face of this slider, facing down towards a collector substrate. The energy provided by the laser pulses produces bubbles at the interface between the metallic layer and the liquid, generating a jet that allows the deposition of the droplets. Using this system, the authors studied HUVECs migration as a function of the distance between patterned cell lines and the presence of HBMSCs. They reported that co-printing HBMSCs reduces HUVECs migration capacity and randomness in the remaining migration events, thus promoting the creation of capillaries. This is of great interest to generate vascularized 3D tissue structures.

Tumor cell migration has been also characterized by X. Wang et al. within a lung tumor-like scaffold bioprinted using a combination of low-temperature molding and syringe-based extrusion of a bioink containing a mixture of A549, 95-D human lung cancer cells and a gelatin-alginate solution [292]. The scaffold was printed as a homogeneous cubic structure formed by grid-shaped layers (Figure 5b). The authors performed cell invasion assays within the scaffold while monitoring MMP2-MMP9 expression, and compared the migration results with those obtained using standard 2D scratch assays. Their analysis showed that the cell invasion and migration capability of A549 and 95-D lung cancer cells is enhanced in 3D structures compared with traditional 2D cultures.

Neural cell behavior in response to bioactive cues has been studied by T.B. Ngo et al. using 3D bioprinted HA-based scaffolds [293]. In one assay, the authors used a methacrylate HA cylinder, printed by extrusion, which acted as a scaffold for testing cell migration (Figure 5c). Schwann cells, a critical component on neural repair and regeneration machinery, were seeded on top of the three cylinders, one made of HA alone, one made of HA mixed with neural growth factors (NGF) and one made of HA mixed with the glial cell-derived neurotropic factor (GDNF). The cylinder surfaces containing the cells were placed facing a glass-bottom dish after culturing seven days to measure cells migration distance. The authors evaluated cell migration under the effects of NGF and GDNF, compared to the control sample. They found no statistically significant differences when comparing control and enriched NGF samples while hydrogel-containing GDNF showed enhanced migration compared to the control HA-only surface.

In summary, we have shown representative examples of the study of cell migration on bioprinted gelatin, HA and collagen scaffolds, to highlight how the use of hydrogels as bioinks can be a powerful tool for studying different biological phenomena, including cell migration. However, bioprinting is a developing field, as fabricating hydrogels with the required rheological properties while allowing cell viability is still a significant challenge. Bioprinting will have a significant impact on biological applications in the near future when the techniques and bioinks mature, allowing researchers to create more complex and functional models to study cell behavior.

### 6.2. Microfluidics

Microfluidic devices have been widely used in research and industry since the birth of this technology in the context of the field of microelectronics [294]. Microfluidic devices consist of micrometer-sized interconnected patterns that can transport and/or hold small fluid volumes in the order of microliters. The device patterns, e.g., cavities, channels, or membranes, are created using microfabrication techniques [295]. Microfluidic devices allow studying complex macroscopic phenomena at the microscopic scale. This has several advantages. On the one hand, the small fluid volumes used allow easier characterization of fluid behavior than when using large, macroscopic scale volumes affected by non-linear effects that complicate experiment characterization [287,288]. Furthermore, the use of controlled flows improves the lattice’s permeability, i.e., oxygen and nutrient supply, compared to classical hydrogels. On the other hand, microfluidics provide cost-effective solutions with high design flexibility, integration and automatization for many application areas [296,297]. In particular, the use of reagents is heavily reduced, which directly affects the cost of the experiments, thus facilitating high-throughput approaches [298].

The development of rapid-prototyping and low-cost fabrication techniques, along with the availability of new biocompatible materials has favored an exponential growth of the use of microfluidic technologies in biomedical applications [295]. Microfluidic devices have been used to study cell behavior -e.g., proliferation or response to mechanical or chemical stimuli- [299], as well as in tissue engineering [300], drug screening [301], and lab-on-a-chip [302] or organ-on-a-chip applications.

Relevant to the scope of this review, 3D microfluidic-based culture models have been used to model cell behavior [303]. In the context of cell migration, the insertion of scaffolds in microfluidic platforms allows mimicking cell to cell and cell to ECM interactions that closely recapitulate those involved in cell migration [304]. Next, describe three recent examples that show the potential of the combination of microfluidics and hydrogels to efficiently study the mechanobiology of cancer cell migration.

Three-dimensional PDMS microfluidic migration devices have been used by M. Anguiano et al., to analyze the role of hydrogel composition and biomechanical properties in H1299 lung cancer cell migration rate and plasticity [42]. The devices, manufactured by replica-molding techniques, and bonded by plasma oxygen to a coverslip glass sample holder, display three parallel channels. The central channel was used to insert a hydrogel of mixed composition-type I collagen and Matrigel at different concentrations- and cancer cells. After polymerization, the hydrogel remains in the channel that becomes a migration chamber. The side channels are used to supply culture medium to the central, polymerized cell-loaded hydrogel. The glass bottom provides a window for microscopic observation with optimal visualization properties (Figure 6a). The authors used quantitative image analysis to show the effect of the ECM composition and geometry in cell migration. Specifically, they reported that pure collagen hydrogels allow slow, mesenchymal cell migration. Mixing collagen with Matrigel at increasing concentrations produces more rigid, heterogeneous hydrogels that favor faster lobopodial cell migration. When a critical concentration of Matrigel is reached, the increased stiffness and heterogeneity of the hydrogel does not translate into a more efficient migration, as the delicate balance existing between the role of FAs for attachment and traction seems to favor attachment versus effective migration tractions.

The migration and alignment of glioma cells towards a flow containing growth factors has been characterized by K.H. Lee et al. using HA-based hydrogels embedded in a microfluidic device [305]. The device, manufactured in PDMS by conventional soft lithography, and bonded to a nanofiber membrane by plasma oxygen, contains a central chamber filled with MMP-sensitive HA embedding glioma cells. The porous membrane located underneath the chamber creates an interface between the gel and a microfluidic channel located below the membrane (Figure 6b). The membrane acts as a diffusion layer while the channel supplies cell media and growth factors. The MMP-sensitive scaffold favors cell scaffold degradation. The authors state that their model mimics cancer tissues with low-flow carrying micro vessels. Using this model, they concluded that under flow-deprived conditions cells remain static and round-shaped, while they adopt an elongated shape and initiate migration when the medium is supplied through the microfluidic channel. Interestingly, cellular orientation after elongation depended on the speed of the flow, being perpendicular to the flow for low flow rates and parallel to the flow for high flow rates. They also reported that cell orientation was dependent of the position of the cells in the hydrogel. Cells located far from the diffusion membrane aligned and migrated perpendicular to the direction of the flow while cells closer to the membrane aligned and migrated parallel to the direction of the flow, thus recapitulating what is seen in tumors.

The neutrophil chemotaxis index has been studied by X. Lu et al. in PDMS-based microfluidic devices by exposing these cells to different lipopolysaccharide gradients [306]. The device, manufactured in PDMS, consisted of two main parallel channels with two different heights, interconnected by five small perpendicular channels. One of the main channels was filled with collagen while the other was used to introduce neutrophils (Figure 6c). The authors demonstrate that neutrophil chemotaxis is reduced by sepsis. Hence, they conclude that neutrophil chemotactic ability may represent an interesting tool to estimate sepsis diagnosis.

There are many other examples of cell migration assays based on the use microfluidic devices loaded with hydrogels. We have shown representative examples that use HA, collagen, and collagen mixed with Matrigel-based hydrogels at different concentrations to illustrate how microfluidics and hydrogels can be efficiently combined to perform relevant biological assays. Specifically, the use of hydrogels embedded in microfluidic devices improves the functionality of these miniaturized devices for modeling cell migration, as the hydrogels recapitulate to a great extent the mechanical and structural properties of the ECM. Microfluidic and hydrogels are two promising areas that will continue growing and allowing researchers to create more complex and functional devices in the future for studying cells behavior.

In summary, bioprinting, microfluidics and in general, biofabrication, are complex, highly innovative processes that tend to be application specific. This is why most of the existing examples of biofabrication have been developed in research environments. However, there are several companies that are developing innovative technologies for biofabrication. Some examples are Envisiontec (Germany), 3D bio-printing solutions (Russia), Regemat (Spain) or Sunp Biotech (China). Other companies, such as Bioink Solution Inc. (Republic of Korea) and CollPlant Biotechnologies (Israel) are working on the development of new biomaterials. There are companies such as Organovo (USA) or ROKIT Healthcare (South Korea) that provide customized solutions for bioprinting tissue models and customized artificial organs, respectively. Hence, biofabrication is slowly being regarded as a business opportunity [307]. The increased investment and business interest will surely lead to significant improvements of the biofabrication technologies. Focusing on the scope of this review article, biofabrication research will surely contribute to improving our understanding of cancer migration, which will translate into a better understanding of the metastatic process and could lead to new personalized treatments and drug-screening methods.

## 7. Conclusions

Cancer cell migration is a complex phenomenon that involves the interplay between many internal, cellular, and external, ECM, biomechanical factors. Studying cell migration in vivo involves the use of animal models and intravital microscopy, which provides very relevant biological information but minimal experimental flexibility. By allowing a minimal reduction in relevance while achieving a great deal of flexibility, 3D hydrogels constitute excellent platforms to mimic the tumor microenvironment and provide a more physiological context than 2D models to study in vitro the principal elements of the mechanobiology of cancer cell migration.

In this review we have covered the most recent technological developments in the fabrication and use of 3D hydrogels for understanding cancer cell migration. In particular, we have discussed the benefits and drawbacks of most hydrogel materials available, with a special emphasis in the implications of using native or synthetic materials, and those of using single-element or mixed composition hydrogels. The choice of the material should be guided by the goal of the study. Single element scaffolds are appropriate to systematically study one or a few aspects of the problem, as they commonly allow very fine tuning of their composition and mechanical properties. Materials of complex composition are more appropriate to study the combination of complex cofactors that cannot easily be isolated, obtaining usually more biologically relevant results, as is the case of disease simulation or drug testing. Indeed, after a revision of the principal elements involved in cell migration we have listed a significant number of studies that provide many novel insights through the use of models based on 3D biomimetic hydrogels. Despite this, current 3D models still remain relatively simplistic and do not fully reflect environmental cues found in vivo, compelling novel fabrication strategies to create scaffolds with potential for clinical translation.

Then, while acknowledging the wide variety of fabrication technologies, we have focused on two of the most recent and important ones, namely the use of bioprinting and microfluidics to create relevant 3D cellular models, and have shown several examples of the use of those technologies, always in the context of better understanding how cancer cells migrate. Altogether, these novel platforms allow comprehensive control of the spatiotemporal interactions between the cell and the matrix, their molecular composition and the presentation of soluble cues, as well as the precise co-culture of different cell types to more closely mimic physiological conditions, which is one of the future challenges to address tumor research.

In summary, we believe that we have shown that 3D hydrogels are novel, extremely powerful assets for understanding the mechanobiology of cancer cell migration, and have provided a significant amount of evidence that novel fabrication technologies, combined with properly selected materials, grant a competitive advantage compared to the complexity and functionality of animal models or simpler 2D cellular models. Based on the success obtained in the past, it is sensible to conclude that future technological developments in biomaterials, combined with bioprinting and microfluidics can only improve the study of this and other biomedical processes, for the benefit of the community, in the form of novel, more relevant and effective cancer-targeted therapies. In particular, having realistic cellular models of cancer, and cancer related events, such as migration, will be instrumental for the scientific community, as scientist will have more powerful models to elucidate the mechanisms of cancer initiation and spread. It should also provide the pharmaceutical industry with tools to test the efficacy on anticancer drugs with a higher chance of success, thus reducing costs that should benefit practitioners and patients in the form of more effective, affordable therapies.

## Figures and Tables

**Figure 1 gels-07-00017-f001:**
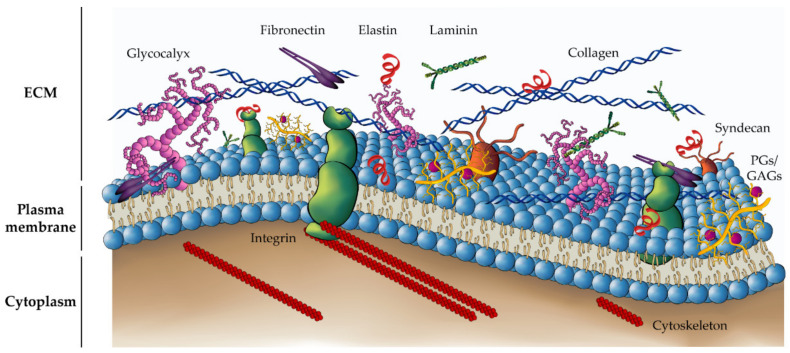
Extracellular matrix (ECM) composition and functions. The ECM is a non-cellular three-dimensional structure made of a large number of cell-secreted macromolecules that provide structural and biochemical support to surrounding cells. Structurally, ECM can be briefly summarized as a blend of water, fibrous proteins, and polysaccharides, e.g., proteoglycans (PGs) and glycosaminoglycans (GAGs). Collagen is the main structural component of the ECM that provides tensile strength to tissues, regulates cell adhesion by anchoring to integrins, and triggers cell differentiation and survival cues. Besides, the collagen lattice interacts with other non-collagenous glycoproteins, e.g., fibronectin, laminin, or elastin- favoring the reinforcement and spatial organization of the ECM. Fibronectin and laminin fibers play a pivotal role in ECM assembly, acting as “adhesive” proteins. Namely, these proteins allow the simultaneous binding to cell-surface receptors, e.g., integrins and syndecans or the cortical glycocalyx-, fibrillar proteins, and other focal adhesion molecules via multiple domains interspersed throughout their structure, which in turn influences cell proliferation, differentiation, and motility. Elastin fibers provide mechanical resilience and elasticity to tissues. Therefore, the collagen/fibronectin ratio confers unique mechanical properties to the tissues that allow both reversible extensibility behavior and the strength to bearing forces. On the other hand, PGs and GAGs fill the interstitial spaces forming a highly hydrated gel by sequestering water molecules, providing compressive strength and buffering properties to tissues. Furthermore, GAGs are a reservoir of growth factors, e.g., TGF-β, EGF, PDGF, etc., that trigger a wide range of fundamental physiological processes ranging from cell proliferation and differentiation to cell adhesion and motility. Indeed, GAGs also modulate cell behavior by interacting with cell-surface receptors to induce cytoskeleton-mediated mechanotransduction of signals and subsequent gene transcription.

**Figure 2 gels-07-00017-f002:**
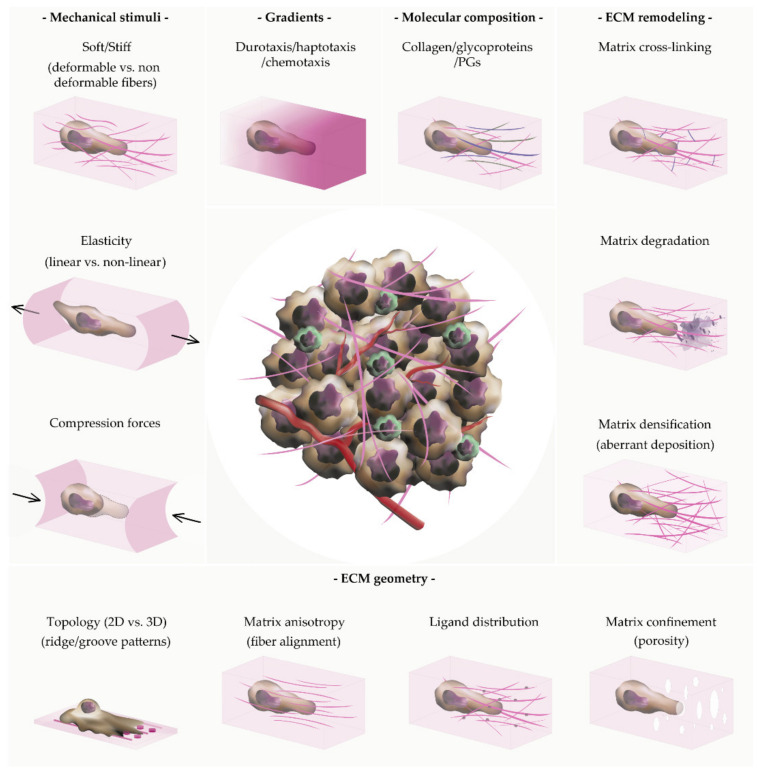
Tumor cell dissemination and metastasis are influenced by the physicochemical properties of ECM and the interactions between cells and matrix. The ECM properties fall into the following categories: molecular composition, mechanical stimuli, ECM geometry, and matrix remodeling. Namely, the content and balance of the ECM constituents -fibrous proteins, glycoproteins, or PGs- determines the topology and mechanical properties of the lattice, as well as the cellular response, via modulation of the surface receptors -e.g., integrins- or the reservoir of growth factors. Mechanical stimuli include different physical cues that affect the deformability of the fibers e.g., strain stiffening effect, the elastic range of the lattice, compressive forces that trigger underlying molecular changes to adapt cell’s morphology to the ECM properties. Thus, actomyosin contractility, focal adhesion (FA) strengthening, and tensile forces adapt dynamically to matrix stiffness and its viscoelastic behavior, promoting tumor cell guidance. Several of these physical cues, as well as soluble bioactive signals contribute to the creation of gradients within the tumor-adjacent stroma e.g., duro-, hapto- or chemotaxis-, which favor directed cell migration. ECM environments are extremely complex being the main variables the dimensionality of the lattice (2D vs. 3D), as well as its topology, understood as the presence and spatial presentation of micro- or nanoscale roughness patterns and/or adhesive sites. The ECM geometry contributes to integrin recruitment, FAs assembly, and the generation of actomyosin-dependent tensile forces, which trigger the alignment of the matrix i.e., anisotropy, necessary for the efficient propagation of tumor cells. Closely interrelated with ECM geometry, matrix confinement involves the presence, spatial distribution, and size of pores within the matrix. The range of confinement is also altered by local remodeling of the ECM through various cell-mediated procedures including, lysyl oxidase family (LOX)-dependent ECM cross-linking, aberrant fiber deposition, or physical rearrangement of the matrix that clumps and align the fibrillar components. These processes modify the mechanical properties of the ECM, especially the stiffness of the matrix, which causes MET reprogramming and tumor progression. Therefore, ECM confinement is one of the major modulators of cellular locomotion, which favors or hinders cellular navigation through 3D environments. Tumor cells often resort to matrix metalloproteinases (MMPs)-mediated pericellular proteolysis to crawl within high-confined environments increasing the metastatic potential of tumor cells. In summary, ECM properties are largely interconnected, and a slight variation in one of them can modify the rest and affect cell behavior. For instance, an aberrant deposition of ECM fibers might increase local stiffness and alter the confinement range, as well as the spatial abundance of ligands, tilting cell response towards invasive phenotypes that rely on tensile displacement and remodeling strategies. A delicate balance of these variables is required to sustain cell and tissue homeostasis and avoid tumor malignancy and tumorigenesis phenomena.

**Figure 3 gels-07-00017-f003:**
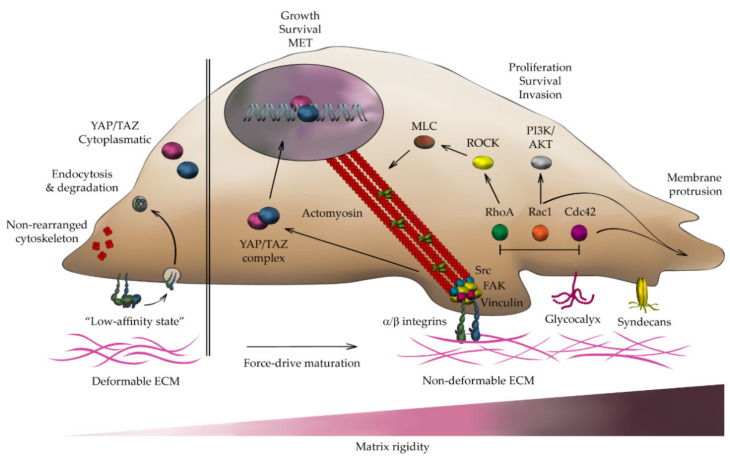
Integrin-dependent adhesion and tensile force mechanotransduction modulate cellular behavior. Cells dynamically probe the biomechanical properties of the surrounding ECM using a host of cell surface receptors including, integrins and syndecans, and activate intracellular signaling pathways accordingly. Soft substrates are not strong enough to activate integrin-mediated signaling and the subsequent mechanotransduction of tensile forces. Consequently, integrins exhibit a low-affinity configuration that hinders either the stabilization of FAs or the actin cytoskeleton rearrangement. In fact, the softness of the matrix elicits the endocytosis and degradation of integrins, enhancing this effect. In contrast, rigid substrates promote integrin recruitment and their effective anchoring to the ECM in a stiffness-dependent manner. Likewise, the cell glycocalyx is compressed by the ECM fibers, which mechanically stresses the integrins, upregulating their activation. Accordingly, integrin activation triggers vinculin coupling and the subsequent activation of the focal adhesion kinase (FAK)/Src complexes that stabilize the FA clutches. Mature FAs bind to actomyosin machinery to sustain the contractile forces required during cell spread. Similarly, syndecan-matrix interaction can activate myosin II to promote cell contractibility and migration in a feedforward positive loop. The mechanotransduction of ECM-derived tensile forces also promotes the YES-associated protein (YAP)/TAZ complex translocation into the nucleus, which triggers the transcription of many oncogenes involved in MET processes. On the other hand, FAK/Src produces downstream signaling events, regulating the activity of Rho GTPases. The crosstalk between RhoA and Rac1 controls the mechanosensitivity of the ECM and provides high cellular plasticity to adapt cell´s mechanobiology to the ECM properties. Indeed, RhoA activates ROCK-signaling concomitant with MLC activation, which stimulates actomyosin contractility and FA assembly. In parallel, the expression of Rac1 and Cdc42 at the leading edge induces the extent of actin-rich membrane protrusions -i.e., lamellipodia and filopodia- promoting directed cell migration. Furthermore, FAK-mediated signaling also promotes pro-survival and invasiveness signals via Rac1/PI3K/AKT pathway. Therefore, the biomechanical properties of the ECM largely determine efficient integrin-dependent adhesion and stress mechanotransduction, which stimulate Rho GTPase signaling and trigger oncogene transcription. Accordingly, these events modulate cell behavior, favoring tumor cell transformation and proliferation, survival, and/or invasion phenomena.

**Figure 4 gels-07-00017-f004:**
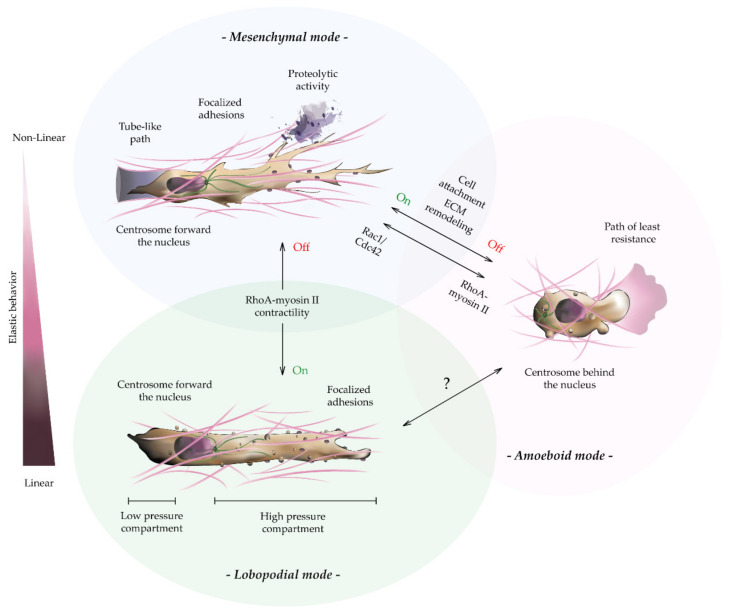
The elastic properties of the ECM and the crosstalk between the small GTPases govern cell migration plasticity within 3D environments. Three modes of 3D cell migration have been described in mammalian tissues, which can be uniquely classified based on the degree of ECM remodeling, cell-matrix attachment, and the modulation of the small GTPases signaling. Mesenchymal-based migration predominates in 3D environments with non-linear elastic behavior. Under these conditions, the polarized signaling of Rac1 and Cdc42 drives the formation of a fan-shaped protrusion at the leading edge by actin polymerization in combination with robust integrin-dependent adhesions. Likewise, Cdc42 signaling locates the centrosome in front of the nucleus facilitating cell axis polarization and directed cell migration. Actomyosin contractility acts throughout the cell body, where it strengthens focal adhesions and increases tensile forces without increasing intracellular pressure, as seen in amoeboid or lobopodial modes. Moreover, mesenchymal locomotion is often accompanied by the degradation of the pericellular area through the secretion of proteolytic enzymes, such as MMPs. ECM remodeling often generates tube-like paths used for collective cancer cell dissemination through dense and poorly organized matrices. On the contrary, lobopodial cells extensively use the actomyosin contractility, focused forward of the nucleus, to migrate in high-confined matrices with a linear elastic behavior. RhoA-myosin II signaling governs intracellular pressure, where the nucleus physically separates the cell body into two compartments. This asymmetrical hydrostatic pressure increases membrane tension at the leading edge, causing numerous, small spherical protrusions along their lateral surface. Despite this, both migration strategies involve elongated shape morphology, strong cell adhesions, and centrosome polarization. Amoeboid-like cell migration is characterized by low-adhesion and matrix remodeling-independent motility. Increased RhoA signaling causes a rapid retrograde flow of myosin II into the leading bleb that allows amoeboid cells to squeeze towards paths of least resistance. Of note, during cancer cell migration, tumor cells often undergo a mesenchymal-amoeboid transition as a consequence of the internal modulation of small GTPAses activity or the external ECM properties. Thus, elongated cancer cells can spontaneously switch into rounded-like amoeboid motility by depletion of MMPs or integrin activity, as well as by migrating through poorly adhesive substrates or upon the modulation of the Rho GTPases signaling. However, there are no clear insights underlying the transition between the lobopoidal and amoeboid mode, thus requiring further experimental work.

**Figure 5 gels-07-00017-f005:**
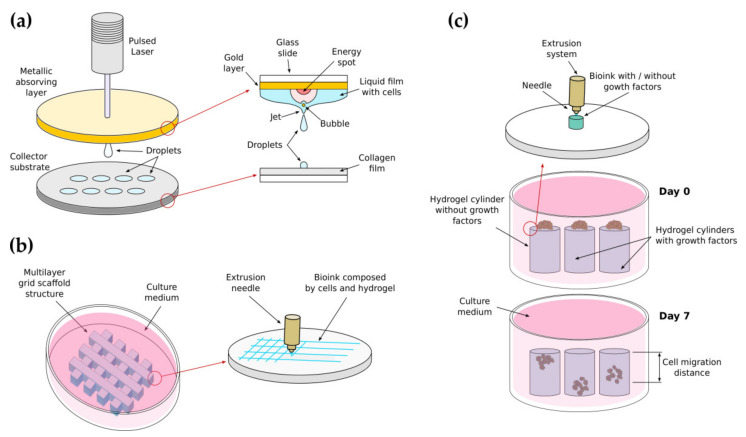
Bioprinted devices for cell migration assays. (**a**) Patterned bioprinted scaffold for cultivating endothelial cells and mesenchymal cells in collagen (Adapted from [291]); (**b**) Grid-shape-cubic scaffold for modeling lung cancer invasion/migration assays (Adapted from [292]); (**c**) hyaluronic acid (HA)-based cylinder used as scaffold for measuring Schwann cells migration capacity under different growth factors effect (Adapted from [293]).

**Figure 6 gels-07-00017-f006:**
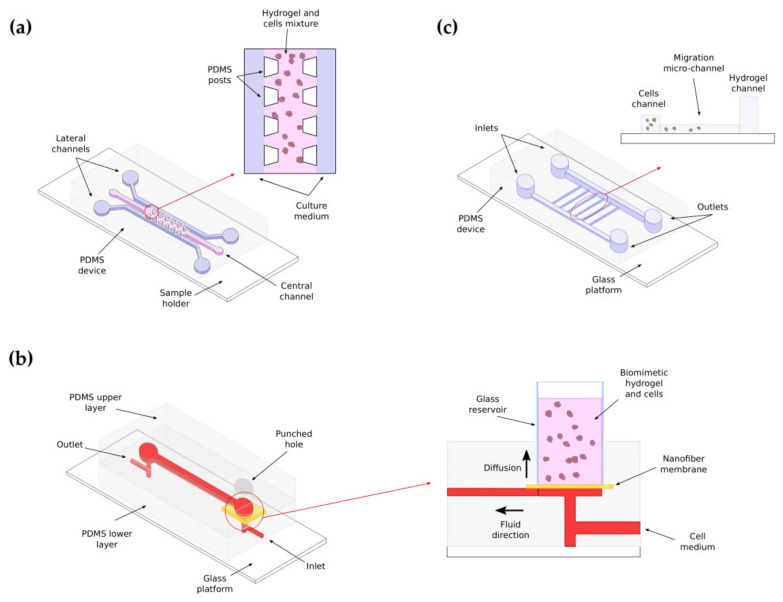
Hydrogel based microfluidic devices for cell migration assays. (**a**) Microfluidic device for studying H1299 lung cancer cell migration capability in collagen and collagen-Matrigel based hydrogel (Adapted from [42]); (**b**) microfluidic device for studying glioma cell alignment and migration capability in hyaluronic acid-based hydrogel (Adapted from [305]); (**c**) Microfluidic device for studying migration effects in neutrophil chemotaxis with a collagen matrix (Adapted from [306]).

**Table 1 gels-07-00017-t001:** Representative overview of scaffold fabrication techniques, materials and applications [179,193,194,195,196,197,198,199,200,201].

Scaffold Fabrication Techniques					
		Advantages	Drawbacks	Materials	Applications
**Classical Methods**	Solvent casting particulate leaching [202,203,204,205,206]	- Highly porous scaffold - Accurate control of porous size and number - Biodegradable - Cost-effective	- Non-uniform porous network - Use of toxic solvents - Limited mechanical properties	- PU, PCL, PEG, PLGA, HA	- Tissue engineering (bone, cartilage)
	Melt molding [207,208,209,210,211]	- Avoid toxic solvents	- Non-uniform porous network - High temperature	- PLGA, PVA, gelatin, chitosan	- Tissue regeneration
	Gas foaming [212,213,214,215,216]	- Highly porous - Controlled porosity	- Limited control of mechanical properties - Poor porous network interconnectivity	- PCL, PLGA, PLA, alginate, gelatin, HA, chitosan,	- Tissue engineering - Drug delivery
	Thermally induced phase Separation [217,218,219,220,221]	- Controlled porous structure - Good mechanical properties	- Only thermoplastics - Irregular size pores - Non-precise scaffold morphology	- PLLA, HApt, PLGA, chitosan	- Vascular scaffolds - Tissue engineering
	Freeze drying [222,223,224,225,226]	- Controlled porous size - No solvent needed - Low temperature	- Small and irregular pore size - Large processing time - Use of cytotoxic solvents	- CMC, Ascorbic acid, chitosan, gelatin, PCL, PLLA, PGA, HA, silk, cellulose, PVA, collagen, HA	- Study cell behavior - Tissue engineering
	Electrospinning [227,228,229,230,231]	- Large surface/volume ratio - Adjusted porosity - Controlled nanoscale fiber distribution	- Limited control of mechanical properties - Pore size - Mechanical stability - Difficult cell seeding	- PLLA, HApt, PCL, PLCL, PGA, PLGA, PEG, EVOH, collagen, gelatin, chitosan, silk	- Drug delivery - Tissue engineering (wound healing, soft tissues, skin)
**Rapid Prototyping**	Stereolithography (SLA) [232,233,234,235,236]	- High resolution - Good pore distribution and control - High porous interconnectivity	- Photopolymerization limits - Massive use of monomers	- PCL, PPF, PLA, PEG, PDMS, HA, chitosan, collagen, gelatin	- Tissue engineering (bone recovery) - Valves reconstruction
	Selective laser sintering (SLS) [237,238,239,240,241]	- Accurate microstructure control - Good mechanical properties	- High operating temperature	- PCL, PLA, PEEK	- Tissue engineering (bones)
	Solvent-based extrusion free forming (SEF) [193,242,243,244,245]	- Accurate microstructures control - High mechanical response	- Extrusions problems (temperature, paste formulation, velocity)	- PCL, PEEK, PEG, PLA, PLGA, PCL, HApt, PDMS, carbon nanotubes, HA, chitosan, alginate, collagen, gelatin	- Cell behavior - Bone recovery - Tissue engineering
	Fused deposition modeling [246,247,248,249,250]	- Accurate microstructure control - Mechanical stability - Fabrication at low temperature	- Limited to biodegradable polymers	- PCL, PPF, PLA, PEEK, PVA, HA	- Tissue engineering
	Bioprinting [251,252,253,254,255]	- Low cost - Structural stability - High geometry complexity - Cell viability - High resolution - Homogeneous cell seeding	- Lack of printable materials - Thermal and mechanical stress to cell	- HA, fibrin, TCP, PLGA, PGA, HApt, PVA, alginate, HA, PEG, nanoparticles, gelatin, methacrylate, PCL	- Cell behavior - Bone recovery - Tissue engineering - Blood vessels - Heart valves - Liver modeling
**Others**	Microfluidics [256,257,258,259,260]	- High resolution - Good pore distribution and control - Functional flows - High-throughput - Multifunctional devices	- Limited interface strength - Integration complexity - Non-standardized devices	- PDMS, PEG, collagen, fibrin, HA, Matrigel, agarose, alginate, gelatin, chitosan	- Organ on a chip - Lab on a chip - Tissue engineering - Drug screening - Study cell behavior

Hyaluronic acid (HA); Carboxymethyl cellulose (CMC); hydroxyapatite (HApt); polycaprolactone (PCL); polyethylene glycol (PEG); polyglycolic acid (PGA); poly-dl-lactic-*co*-glycolic acid (PLGA); poly-l-lactic acid (PLLA), polyvinyl alcohol (PVA); polystyrene (PS); polyvinylpyrrolidone (PVP); poly-lactic acid (PLA); polyurethane (PU); polydimethylsiloxane (PDMS); polyether-ether-ketone (PEEK); polypropylene fumarate (PPF).

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
