# Peer review of "Modeling the Mechanobiology of Cancer Cell Migration Using 3D Biomimetic Hydrogels"

_gels, 2021, doi:10.3390/gels7010017_

Round 1

Reviewer 1 Report

The topic is new, futuristic and no similar thematic review articles have been published till date. Does correlate and give valuable information on Three-dimensional hydrogel-based cellular models of the mechanobiology especially in cancer cell migration. However, extensive English language revisions are required. The scrutiny does add significant insights in the respective field of research. Further, few minor revisions are required in order for the review to be accepted for publication.

1.      Title should be repharsed more formally. Abstract lacks highlighting why studies on cancer cell migration are important.

2.      The contents in the tables needs to be improved.

3.      The discussions are not example based does not connect properly with just directional explanation. Authors are advised to add certain relevant examples so that readers get a detailed information.

4.      A table listing the industries applying these methodologies is recommended for the review.

5.      The review lacks upbringing the importance of biomaterials and their importance in highlighting to build and use for the cancer cell migration. New references suggested below should be included in the discussion.

6.      A new section on listing the biomaterials and their fabrication for 3D hydrogel matrices should be added.

7.      A detailed explanation on how this correlation is important for different stakeholders such as healthcare professionals, patients, scientist and CRO needs to be commented in the conclusion section.

8.      It would be great if authors can interrelate these concepts on upbringing or fasten the vaccine production or better understanding the COVID-19. A good reasoning would be helpful to readers.

9.      Also below are the few latest 3D hydrogel based models as delivery systems developed and evaluated for the same, authors are strongly advised to cite and mention in the respective discussion sections. Authors are strongly suggested for these recommendations to be considered for further evaluation.

Materials Science and Engineering: C119, p.111460.

Processes. 2021 Jan;9(1):45.

Marine Drugs. 2020 Apr;18(4):201.

Critical reviews in biotechnology. 2016 May 3;36(3):553-65.

Biomaterials. 2016 Mar 1;81:72-83.

AAPS PharmSciTech20(7), p.297.

Biosensors and Bioelectronics. 2020 Jan 1;147:111757.

Current Drug Delivery. 2016 Mar 1;13(2):211-20.

Scientific reports. 2018 Mar 28;8(1):1-2.

Tissue Engineering Part C: Methods. 2016 Jul 1;22(7):708-15.

Author Response

Please find our response in the attached Word file.

Reviewer 2 Report

In this manuscript, Morales et al. reviewed three-dimensional hydrogel-based cellular models for cancer mechanobiology and cell migration.

The review provides a brief overview of cell migration, its importance and consequences in cancer mechanobiology, and the role of extracellular matrix components. Then a brief overview of the extracellular matrix and materials used to mimic three-dimensional ECM scaffolds are discussed with representative examples. The authors provided extensive literature reports with bio-based ECM constructs and synthetic 3D scaffolds, their mechanical behavior, and their relevance in cell-matrix interactions, migration, and proliferation. 

Finally, emerging techniques such as 3D bioprinting and microfluidics to study cancer migration and metastasis are discussed.

Relevance and importance: There is has been tremendous progress in recent years to develop 3D hydrogel-based ECM mimicking systems to study cancer cell culture, tissue culture, organdies, and mimic metastasis. Numerous efforts have been made to overcome some of the unwanted cell-matrix interactions in the culture system (from unknown growth factors) and control the ECM components' mechanical properties. Such 3D systems are clinically relevant as they provide good in vitro models to establish drug response for various cancers systematically. 

Therefore, this review is timely and will be of interest to a diverse group of readers.

Comments/Corrections: 

  1. A Figure or schem showing ECM and its components will be of benefit to the readers. 

Author Response

Please find our response the the reviewer's comments in the attached Word file.
